# Iterative Spatial Crowdsourcing in Peer-to-Peer Opportunistic Networks

**Jurairat Phuttharak [1],\***  **and Seng W. Loke [2]**

[1]  Department of Management Information Technology, Prince of Songkla University, Trang Campus, Trang 92000, Thailand

[2]  School of Information Technology, Deakin University, Geelong 3216, Australia; seng.loke@deakin.edu.au

\*  Correspondence: jurairat.b@psu.ac.th

**Abstract:** Spatial crowdsourcing is a potentially powerful method for incorporating human wisdom into mobile computations to solve problems while exploiting the advantages of mobility and context-awareness. This paper proposes and investigates task assignments and recruitment in iterative spatial crowdsourcing processes to find regions of particular interest among a collection of regions. We consider cases where associations between regions can be exploited to reduce costs and increase efficiency in crowdsourcing. We describe five approaches, incorporated into crowdsourcing algorithms, for reducing the cost (the number of queries required) and increasing the efficiency (reducing the number of rounds of querying required) in using such spatial crowdsourcing. We demonstrate the performance improvements gained using these approaches based on simulation scenarios. The findings show the interplay and relationships among our proposed approaches using a range of metrics including responses, energy consumption, costs, and time usage. These metrics are demonstrated via a range of scenarios, showing that our proposed approaches can lead to improved performance over randomly choosing regions for inquiry.

**Keywords:** spatial crowdsourcing; mobile crowdsourcing; spatial finding; P2P crowdsourcing

---

## 1. Introduction

In the past decade, the emergence of the crowdsourcing paradigm has dramatically changed the landscape of business by building better products and services. Crowdsourcing, simply known as the power of the crowd, involves seeking knowledge, goods, or services from a larger group of people. These people submit their ideas in response to online requests made through social media, smartphone apps, or dedicated crowdsourcing platforms [1–4]. Recently, the rapid adoption of mobile devices has provided a new data collection paradigm, often termed as mobile crowdsourcing. Mobile crowdsourcing, i.e., crowdsourcing through mobile users, presents significant new opportunities and challenges, with enormous possibilities for human computations and task dissemination with spatial and temporal properties [5–10].

Due to the pervasiveness of mobile devices and their superb functionality, workers can perform a set of spatial tasks (i.e., tasks related to a geographical location and time) posted by a requestor. These tasks can be quickly and easily completed by mobile device users (e.g., crowdsourcing for car park spaces, locations of crowds, maps of areas, transport demand, emergency needs, photos/video at different locations in a parade, location of flora and fauna). This is different from traditional approaches wherein workers in spatial crowdsourcing are required to perform a set of tasks by physically travelling to, or being present at, certain locations at particular times. The tasks accomplished by such crowdsourcing may be performed over extended periods of time providing data for analytics, or in an ad-hoc real-time on-demand manner. An example of this is Postmates, a company offering

on-demand food and delivery, which is available all around the US. In addition, Uber, Grab, and Lyft are successful platforms for crowdsourced-taxi services where tasks are allocated to workers based on their availability and spatio-temporal requirements.

Incentives are key to the success of crowdsourcing applications. This motivates people to contribute to a crowdsourcing effort. The costs incurred may be related to monetary payments for answers or expenses, i.e., paying for contributions, and time efficiency, e.g., the time required to achieve an adequate response. Implementing crowdsourcing applications is still viewed as challenging because the systems largely rely on humans. Therefore, the process might take longer in order to find workers to respond to queries and to collect results from the tasks that are completed. Therefore, recruitment and motivation techniques under a limited budget are essential components for developing crowdsourcing systems. Algorithms where humans are viewed as data processors have been explored for determining the maximum [11], for filtering [12], and for finding a subset of items with a given property among a given unstructured collection [13], while taking into consideration the need to optimize cost and efficiency.

In view of these challenges, we propose and investigate task assignment/recruitment in spatial crowdsourcing processes to find regions of particular interest from among a collection of regions where the potential contributors are in a peer-to-peer network.

Typically, in the context of mobile crowdsourcing, the workers for crowdsourced queries are people within the area with mobile devices. Therefore, the queries posed to them and their answers have spatial properties. In this research, we consider cases where associations between regions can be exploited to reduce the costs and increase efficiency in crowdsourcing. Often, information about a region provides clues about its neighboring regions: a region that is crowded might be adjacent to another crowded region or a polluted region might be adjacent to another polluted region, although this is not always the case. We argue that this is the case for a number of real-world phenomena including car parking, 3G/4G bandwidth, crowded areas, and noise pollution, as noted in [14]. So, for example, if one wants to use crowdsourced queries to find regions where car parking is available, regions where there is currently high 3G/4G bandwidth, regions which are crowded, or regions with noise pollution above a set threshold, then neighborhood, or proximity, associations can be exploited.

In the rest of this paper, we first review related work in the "Related Work" section. Then, we outline the spatial finding problem and discuss possible solutions in the "Problem Overview" section. We introduce five approaches for spatial finding in peer-to-peer (P2P) crowdsourcing networks in the "Task Propagation Strategies in P2P Spatial Crowdsourcing" section. The simulation and analytical results for spatial crowdsourcing on P2P opportunistic networks are described in the "Simulation and Evaluation Results" section. We look at the importance and applicability of our results in the "Discussion" section. We then conclude with the "Conclusion" section.

## 2. Related Work

Due to the rapid development of mobile networks and the widespread usage of mobile devices, Spatial Crowdsourcing has become a promising research area. In fact, spatial crowdsourcing has stimulated a series of recent industrial successes, including P2P ride-sharing services (e.g., Uber (https://www.uber.com [Last accessed 25 June 2020]) Lyft (https://www.lyft.com [Last accessed 25 June 2020]) and Grab (https://www.grab.com [Last accessed 25 June 2020]), citizen sensing services (e.g., Waze (https://www.waze.com [Last accessed 25 June 2020]) and OpenStreetMap (https://www.openstreetmap.org [Last accessed 25 June 2020])), product placement checking in supermarkets (e.g., Gigwalk (https://gigwalk.com [Last accessed 25 June 2020]) and TaskRabbit (https://www.taskrabbit.com [Last accessed 25 June 2020])) and real-time Online-To-Online (O2O) services (e.g., Instacart (https://www.instacart.com [Last accessed 25 June 2020]) and Postmates (https://postmates.com [Last accessed 25 June 2020])). Despite the success of these platforms, because of the unique spatio-temporal dynamics in spatial crowdsourcing, the designs of theories and systems still need to improve. Our focus

here has been on information crowdsourcing, where queries are used to develop real-time maps of regions that satisfy specific properties.

Recently, there have been research surveys in [4,8,15] discussing the core issues of spatial crowdsourcing (SC), including task assignment, incentive mechanism, privacy protection, the absence of real-world data sets, scalability, and quality of reported data. One of the major challenges with SC is task assignment [16,17]. Hein [16] categorized task assignment in spatial crowdsourcing into static (offline) and dynamic (online) scenarios. In static scenarios, most efforts were put into maximizing the total number of valid assigned pairs (tasks and workers) [18–21]. Meanwhile, online scenarios aim to maximize the number of assigned worker's task pairs under a budget constraint where workers appear dynamically on platforms. Existing solutions often develop two-sided online matching algorithms to adapt to subsequent unknown arrival objects [22–25]. Privacy and trust issues in spatial crowdsourcing are important for protecting workers' privacy and validating the results provided by workers. Recently, many related approaches have been proposed to cope with location privacy issues for this type of crowd wisdom [26–29]. These methods address privacy by masking the location information based on a differential privacy approach [27,29].

The advent of mobile and wireless communication has shown that the universe of distributed computing is indeed much wider than previously thought. Several approaches in [30–32] have been introduced for distributed computational environments, such as mobile sensor networks, delay tolerant networks, and mobile ad hoc networks. In opportunistic networks, the designing efficient forwarding schemes is one of the main challenges. Most proposals use centrality as their routing metrics, which requires certain knowledge of the network's configuration and topology. Many opportunistic routing protocols have been proposed in the literature [30]. Epidemic routing protocol [33] is a basic routing protocol which employs the flooding technique for delivering messages. Whenever a node encounters another node, it transfers all the messages which the other node doesn't already possess. History Based Routing Protocol [34] is a context-based routing protocol that uses historical data and lists of previous intermediaries to prioritize the schedule of packets transmitted to other peers and the schedule of packets to be dropped. In Spyropoulos et al. [35], the authors proposed ProPHET, a probabilistic routing protocol that makes forwarding decisions based on the computed delivery predictability of intermediate nodes.

A routing protocol in this context refers to the propagation strategy for disseminating crowd tasks among peers, which makes it a significant aspect of data transmission in mobile crowdsourcing systems. Mobile users can easily interact with each other in a mobile network fashion, which can be regarded as an ad-hoc network, supporting multi-hop routing, content forwarding, and distributed, decentralized processing. Phuttharak and Loke [36] investigated task propagation models devised to support mobile crowdsourcing in intermittently connected opportunistic networks. They applied the k-walker random walk technique for distributing tasks over a geographical area. In their approach, a task of the source node was distributed to others with no more than *n* degree within a communication range (e.g., Bluetooth, WiFi-Direct). The study simulated the distribution of crowd tasks in mobile crowdsourcing networks with limited communication ranges and explored the factors that impacted crowd task propagation and energy usage for each node.

Moreover, the work in [37] proposed a framework called SmartOpt for searching objects (e.g., images, videos, etc.) which were captured by users in a mobile social community. This platform exploited the location data made available by the crowd to optimize a searching process with peer-to-peer systems. This approach is able to minimize energy consumption during searching, reduce the query response time when the search is conducted, and also maximize the recall rate of the user query. Li et al. [38] proposed a software-defined opportunistic network structure for mobile crowdsensing. This framework provided fine-grained management of the opportunistic network and recorded user contributions in data forwarding through accurate statistics. The study in [39] extended the sensing capability of smartphones by allowing them to identify their nearest geographic neighboring nodes in real-time in a process called CrowdCast. This framework is beneficial to the

crowdsourcing paradigm since it provides full access to the mobile workforce and adds the temporal dimension to location data in order to exploit trajectory-related information. We have reviewed a range of methods for crowdsourcing using centralized and decentralized architectures in [10]. Our review suggests that our approach is novel and little work has attempted to compare the algorithms for peer-to-peer crowdsourcing as we do in this paper.

In earlier work on task assignment in spatial crowdsourcing, Tong et al. [40] proposed algorithms with theoretical guarantees to maximise the total utility score of the assignment. Tao et al. [41] recommended routes for workers to maximize the total utility. Zeng et al. [42] assigned tasks to workers while trading off quality and latency of task completion. Tong et al. [25] proposed a match-based approach to solve the dynamic pricing problem in spatial crowdsourcing. Gao et al. [43] recommended top-k teams to minimize the total cost of the recommended teams. Different from such work, our work here has considered the complexity of crowdsourcing over P2P networks where energy consumption (not just for each device but the network as a whole) is considered in addition to cost and time efficiency considerations. We have also introduced five approaches for P2P crowdsourcing over networks and compared these five approaches based on cost, battery usage, and efficiency metrics. While they are based on adaptations of previous algorithms, our multi-criteria approach we develop in this study is novel.

## 3. Problem Overview

The purpose of spatial crowdsourcing (SC) is to require workers who are physically at specific locations at a particular time to complete the tasks. One of the major challenges with SC is task assignment. This deals with allocating a specific set of crowdsourced tasks based on spatio-temporal requirements to the set of crowd workers who can potentially finish these tasks most accurately and efficiently. For example, if we want to find at least $k$ regions with available car parking spaces, a large area is divided into a set of regions. We then can ask the crowd about parking availability in each region, assuming that the task assigned to answer the (location-dependent) query is only possible for people at the given location at the time of the inquiry. Each time we ask the crowd about a region, we assume that we incur a cost. There are different optimization goals to achieve to deal with the problems of task assignment described in [9], for example, maximizing the number of assigned tasks, maximizing the number of tasks completed by workers before a deadline, minimizing the incentives paid to the workers, and maximizing the rewards received by the workers.

Based on the above problems, several SC applications come with a centralized approach which uses web-based and client-server communications that provide access via either conventional smartphones or workstations. However, mobile ad-hoc network communication involving multi-hop transfers and decentralized processing is increasingly interesting in mobile applications today. In mobile ad hoc networks, standard wireless communication technology, such as traditional WiFi infrastructure networks, WiFi direct, Bluetooth and LTE direct, supports these kinds of connections. Suppose that we wanted to find a not-so-crowded café, answers being given by people near or within the region. We could obtain this information by means of peer-to-peer (P2P) interactions. Mobile users could check in, share nearby information (photos, news, or deals) with friends in a mobile P2P manner. Indeed, it is also possible for mobile users to discover friends in nearby parks, or to arrange meet-ups and conferences, via mobile P2P collaborations or even to find a high bandwidth (WiFi, 4G, or otherwise) region. P2P networking enables P2P networks to form, thereby providing a basis for P2P spatial crowdsourcing, which we explore in this paper.

The original version of SC as defined in [14] does not deal with P2P spatial crowdsourcing properties and networks as we do in this paper. We define a P2P SC version of the above problem, and while exploring the above solution ideas, we consider P2P spatial heuristics for picking who to ask. The general notion of cost-latency trade-offs, however, is also considered here.

*3.1. Our Problem*

Assuming that a large area $R$ is partitioned into $n$ regions $\{r_1, ..., r_n\}$, the problem is to find a set $S \subseteq R$ of at least $k \leqslant n$ regions, each of which evaluates to true for a given predicate F representing set criteria, i.e., $F(r) = TRUE$, for each $r \in S$. We also want to solve this problem with the lowest total cost for each of the queries/tasks (assuming we need to pay to get a question about a region answered), while reducing battery usage on mobile devices, and in the most efficient way (the number of answers and the number of rounds of questions required). Moreover, the queries are to be issued over a peer-to-peer network (e.g., a collection of mobile devices and/or vehicles).

*3.2. Metrics*

There are three factors to deal with in any solution to the problem.

Cost—In reality, each P2P spatial crowdsourcing operation has a cost, e.g., monetary costs for payments for answers. In monetary costs, we assume that each time a query is issued to find out something about a region, there is a related cost. A cost $\phi$ includes the cost of issuing the query, as well as incentives paid for answers to a query. A cost $\phi$ is incurred so that if we ask about $k$ regions, we incur a total cost of $k.\phi$.

Region response—Although we seek to reduce the cost of the query, we must ensure that our techniques do not degrade the worker experience. Region response is measured by the following metrics:

- Number of responses—this refers to the size of the total result set for each region. In a P2P spatial crowdsourcing network, a node will disseminate a query in the region through mobile peers in its range and then obtain responses from the mobile device peers. In general, the number of responses would be less than or equal to the number of peers receiving the query.
- Number of rounds required—this is defined as the number of rounds of querying required. In each round, a set of queries is issued in parallel to find out about a particular chosen set of regions (assuming one query per region). Note that we use the term "iterative" to refer to the processes we are studying in this paper, each of which is comprised of one or more rounds of queries (that is, there is an iteration till the number of regions of interest has been found).

Energy consumption—The energy consumption of mobile devices has increasingly become a concern in spatial crowdsourcing because of limited battery life. Moreover, power usage is one key factor affecting workers' willingness to participate in spatial crowdsourcing tasks. Energy is consumed in all aspects of its application, ranging from sensing and processing, to data transmission.

We assume that the battery level of a peer's device is spent during two processes: a run-time process (the processes when the node sends, forwards, or receives a worker's tasks) and an idle-time process (the time spent waiting for worker responses). In this study, the total energy consumption in the P2P spatial crowdsourcing network was measured to compare the overall performance of each scheme implemented.

## 4. Task Propagation Strategies in P2P Spatial Crowdsourcing

In this section, we describe the main characteristics of our propagation strategies in P2P spatial crowdsourcing. This refers to a distributed problem-solving model for finding regions of interests using mobile crowdsourcing. Mobile users can interact with each other throughout a collection of regions in a mobile network supporting multi-hop routing, content forwarding, and distributed decentralized processing. Our solutions to the above problems are inspired by the solutions from the seminal work in [44], which was initially developed to search for particular data items stored in P2P data-centric systems.

We present five approaches for efficient spatial finding in P2P crowdsourcing networks, including (1) flooding (baseline), (2) Random-X-Walker, (3) Random-X-Walker with $\gamma$, (4) Multi-Criteria Neighbor Selection, and (5) Multi-Criteria Neighbor Selection with $\gamma$.

Notably, the flooding approach is exploited as a simple scheme for the search technique in P2P networks shown in [45]. We have adjusted this algorithm in our current study to suit spatial P2P crowdsourcing environments. This approach is treated as a baseline algorithm to compare the effectiveness of our proposed methods. We describe these approaches in details in the following subsections.

The network topology investigated in this study is similar to [36], which is based on spatial random node distributions and presumed short-range wireless connectivity between the nodes shown in Figure 1. Nodes (N) are assumed to be randomly placed in an area/region (A) where $A = \pi r^2$. A constant node density (the expected number of nodes per region unit) is defined as $\rho = N/A$. In our case, a requestor propagates queries to workers whom it discovers. When receiving the queries, a worker answers the queries and replies to the requestor. Moreover, the worker is able to forward tasks (in effect, the queries) to others. This process might continue with the subsequent peers. We assume that a node can act as both a requestor and a worker in this network model. During the propagation process, time-to-live values are defined to limit the lifetime of queries so that the action of forwarding queries to others would eventually be stopped and the responses returned along the path through which the queries were propagated. Each subsection which follows is structured as follows: for each algorithm, we present the idea, the algorithm's pseudo-code and an analysis of the algorithm.

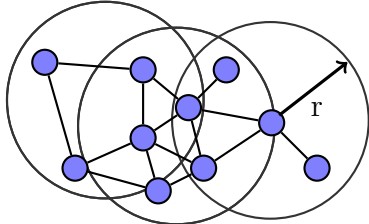

**Figure 1.** Modelling the topology of an ad-hoc network.

### 4.1. Flooding

### 4.1.1. Idea

The most popular query routing algorithm in P2P networks is based on the flooding scheme [45]. In the flooding-based search technique, a querying peer node sends a search query to all its neighbors without any constraints. This scheme has been applied to spatial crowdsourcing contexts. In our case, a peer sends the query to all of its neighbors. Then, each peer forwards the query to all of its neighbors except the peer who originally sent the query. The process continues with the subsequent peers until the expiry time of the query. Each query starts with an initial time-to-live (TTL) value, and, when the query forwards to another node, the TTL value is reduced across the queries. To avoid the problem that one peer receives the same query several times, a unique number has been set for each query. If a peer has been sent a query with the same number as a previously received query, the new query is discarded.

### 4.1.2. Algorithm

We assume that each region is an item responding with a binary answer YES or NO (TRUE or FALSE). We developed the algorithm below to find particular regions using a P2P network, given a set of regions $R$ and $\alpha$ which denotes the fraction of regions (we call positive regions) of $R$ where $F$ evaluates to TRUE (and the rest of the regions, $F$ evaluates to FALSE, and we assume that $\alpha > 0$). Additionally, assuming that we are finding $k$ positive regions from $R$, where there are at least $k$ regions that can satisfy $F$, i.e., $k \leq |R| \cdot \alpha$. The flooding scheme is represented in Algorithm 1.

Algorithm 1 terminates when either $k$ regions satisfying $F$ are found or when the time ($\tau$) of query distribution has expired. $\tau$ is the value that limits the lifetime of the query propagation. Every query is

tagged with a $\tau$ value which is reduced after the node forwards the task to others in each iteration. The following equation (described in [36]) is used to calculate $\tau$ for tagging the query in the current node before forwarding it to others. Note that the minimum value of $\tau$ should be greater than or equal to $\mu$ of each device.

$$\tau = T_{source} - T_{forwarder} - \mu \tag{1}$$

While we have not found enough regions (and there are still regions from $R$ that are not yet observed), we iteratively select a subset of regions to observe according to the function selectNeighbors$(S, k, D, R \setminus O)$ shown in Algorithm 2. In Algorithm 1, askCrowdAbout$(N)$ issues $|N|$ queries in parallel to ask about regions in $N$.

---

**Algorithm 1:** Flooding $(k, F, R, \tau, \mu)$

---

**Input** ：$k$: number of regions to find, $F$: predicate on region, $R$: set of regions,
　　　　$\tau$: Time-to-Live, $\mu$: time for transmitting a query,
　　　　$N$: set of selected unobserved neighbors, $S$: set of source nodes

**Output**：$D$: set of found regions, $O$: set of observed regions

$\quad D := \varnothing$　　　　　　　　/* initial value for found regions */
$\quad N := \varnothing$　　　　　　　　/* initial value for unobserved neighbors */
$\quad S := \{Requestor\}$　　/* initial value for source node equal to "Requestor" node */
$\quad \tau_i := \tau$　　　　　　　　　/* expiry period of query of $S$ */

$\quad$**while** $(|D| < k)$ AND $(\tau_i > \mu)$
$\qquad N :=$ selectNeighbors$(S, k, D, R \setminus O)$;
$\qquad$askCrowdAbout$(N)$;
$\qquad (A \cup B) :=$ response from crowd about $N$, where
$\qquad\qquad A := \{r \in N|\ F(r) = TRUE\}$ and $B := \{r \in N|\ F(r) = FALSE\}$
$\qquad\qquad$and $(A \cup B) \subseteq N$;

$\qquad S := N$;　　　　/* assign the neighbor nodes to source nodes */
$\qquad \tau_i := \tau_i - \mu$;
$\qquad D := D \cup A$;
$\qquad O := O \cup N$;
$\quad$**end**
**return** $D$ and $O$;

---

---

**Algorithm 2:** selectNeighbors$(S, k, D, R \setminus O)$

---

**Input** ：$S$: set of source nodes = $\{s_i, ..., s_n\}$, $k$: number of positive regions to find,
　　　　$D$: set of found regions, $R$: set of regions, $O$: set of observed regions,
　　　　$n_i$: set of selected unobserved neighbors of $s_i$

**Output**：$N$: set of selected unobserved neighbors to be queried

$\quad N := \varnothing$; /* prepare empty list for collecting new neighbors from source nodes (S)*/
$\quad c := k - |D|$;
$\quad$**for** each $s_i$
$\qquad n_i :=$ select all neighbors in the range of $s_i$ from $R \setminus O$, where $|n_i| \leqslant c$;
$\qquad N := N \cup n_i$;
$\quad$**end**
**return** $N$;

---

### 4.1.3. Analysis

If the quality of the results in this system is measured solely by the number of responses, then the flooding technique is ideal because it sends the query to every possible region (i.e., all nodes in the network) as quickly as possible. However, if efficiency (and use of resources) is the metric of choice, the flooding scheme wastes resources and is more expensive as mentioned earlier. To minimize total costs, we can ask about one region at a time (in any order) and stop inquiring whenever $k$ regions are found to meet the criteria, so that we never ask more questions than required or get more than $k$ positive answers in the time $\tau$. In the worst case, this scheme can lead to $|R|$ rounds. To manage the trade-offs between cost and efficiency, we propose and investigate two other possible solutions which will be described in the next section.

### 4.2. Random-X-Walker

#### 4.2.1. Idea

Random-X-Walker (RXW) has been proposed to mitigate the consumption of resources and reduce costs related to the flooding technique. It has been adapted from our previous work in [36]. In this algorithm, a requestor initiates a search for regions by forwarding a query to $X$ neighbors chosen randomly (using a uniform probability distribution), instead of forwarding the query to all available neighbors as in the flooding algorithm. The process continues until sufficient regions are located and each peer selects the next $X$ neighbors to randomly forward the query. This random walker mechanism executes until the $\tau$ value expires or it receives $k$ regions responses. In Random-X-Walker, each query has a $\tau$ value just like in the flooding scheme. The value is decreased every time it travels between regions and the query is forwarded as long as its $\tau$ value still is greater than $\mu$ (the time for transmitting a query/packets for each peer/device). Moreover, in this technique, we also use a maximum hop distance set by the requestor in order to limit the forwarding of the query. Hop distance is the number of peers forwarding the query from node to node. We modified this approach from Algorithms 1 and 2. Therefore, the following Algorithms 3–6 show how the Random-X-Walker works.

Limiting the forwarding of the query using distance reduces costs and increases efficiency in spatial crowdsourcing. However, this approach still encounters an issue when the total number of positive regions found does not reach the number of regions required at that time. A solution for this problem might be allowing the requestor to initiate multiple rounds of questioning, even if it would take longer. This technique is also shown in Algorithm 3 and Figure 2.

The policy for propagating the query in multiple rounds is described as follows. Firstly, the number of regions required is initialized by a requestor. Then the hop distance would be calculated using Equation (3) in the function calMaxHops($D, k, x$). For example, the requestor assigns the number of regions as $k$ regions, the hop distance can be calculated as "a" hops, and then queries are sent randomly to neighbors within "a" hops. When a node at depth "a" receives and processes the query, it will store the query temporarily (and not forward it). The query therefore becomes frozen at all nodes at "a" hops from the requestor. Meanwhile, the requestor will receive the responses from the regions that have processed the query. If the number of positive regions has been satisfied, then the algorithm terminates; otherwise it will start the next round, initializing a depth "b" (re-calculating the hop distance from the number of found regions, shown in Algorithm 3). In the next round, only one node at depth "a" is selected randomly (calling the function selectOneNeighbor($N$)) that will forward the query along its path within "b" hops. The process continues in a similar fashion to other levels in the network until the $\tau$ value expires or the number of regions required is obtained.

---

**Algorithm 3:** Random-X-Walker $(k, F, R, \tau, \mu, HL, x)$

---

**Input** : $k$: number of regions to find, $F$: predicate on region, $R$: set of regions,
　　　　$\tau$: Time-to-Live, $\mu$: time for transmitting a query, $x$: node degree,
　　　　$H$: current hop level, $HL$: maximum number of hops that a query can be propagated

**Output:** $D$: set of found regions, $O$: set of observed regions, *round*: number of rounds

$$D := \varnothing \qquad \text{/* initial value for found regions */}$$
$$N := \varnothing \qquad \text{/* initial value for unobserved neighbors */}$$
$$S := \{Requestor\} \quad \text{/* initial value for source nodes */}$$
$$\tau_i := \tau \qquad \text{/* expiry period of query of } S \text{ */}$$
$$H := 0; \qquad \text{/* initial current hop level */}$$
$$round := 0; \qquad \text{/* initial round */}$$

**while** $(|D| < k)$ AND $(\tau_i > \mu)$
　　　　$HL := \text{calMaxHops}(D, k, x);$
　　　　**while** $(H < HL)$ AND $(|D| < k)$ AND $(\tau_i > \mu)$
　　　　　　　$N := \text{selectNeighbors}(S, k, D, x, R \setminus O);$
　　　　　　　$\text{askCrowdAbout}(N);$
　　　　　　　$(A \cup B) := $ response from crowd about $N$, where
　　　　　　　　　　$A := \{r \in N \mid F(r) = TRUE\}$ and
　　　　　　　　　　$B := \{r \in N \mid F(r) = FALSE\}$ and $(A \cup B) \subseteq N;$
　　　　　　　$S := N;$
　　　　　　　$\tau_i := \tau_i - \mu; H + +;$
　　　　　　　$D := D \cup A;$
　　　　　　　$O := O \cup N;$
　　　　**end**
　　　　$S := \text{selectOneNeighbor}(N);$
　　　　$H := 0; round + +;$
**end**
**return** $D$, $O$ and *round*;

---

**Algorithm 4:** calMaxHops$(D, k, x)$

---

**Input** : $D$: set of found regions, $x$: node degree
　　　　$k$: number of positive regions to find

**Output:** $HL$: maximum number of hops

$$c := k - |D|;$$
$$HL = \left\lceil \frac{\ln\left(\frac{c(x-1)}{x} + 1\right)}{\ln(x)} \right\rceil;$$

**return** $HL$;

---

**Algorithm 5:** selectOneNeighbor$(N)$

---

**Input** : $N$: set of selected unobserved neighbors to be queried

**Output:** $S$: set of source nodes

$$S := \text{randomly selected member from } N;$$

**return** $S$

---

**Algorithm 6:** selectNeighbors$(S, k, D, x, R \setminus O)$

---

**Input**  : $S$: set of source nodes = $\{s_i, ..., s_n\}$, $k$: number of positive regions to find,
$D$: set of found regions, $R$: set of regions, $O$: set of observed regions,
$n_i$: set of selected unobserved neighbors of $s_i$, $x$: node degree

**Output**: $N$: set of selected unobserved neighbors to be queried

$\qquad N := \varnothing;$
$\qquad c := k - |D|;$
$\qquad$**for** each $s_i$
$\qquad\qquad n_i :=$ randomly select $x$ neighbors within the communication range of $s_i$ from
$\qquad\qquad R \setminus O$, where $|n_i| \leqslant c;\qquad (n_i := c$ neighbors, when $x > c)$
$\qquad\qquad N := N \cup n_i;$
$\qquad$**end**
**return** $N;$

---

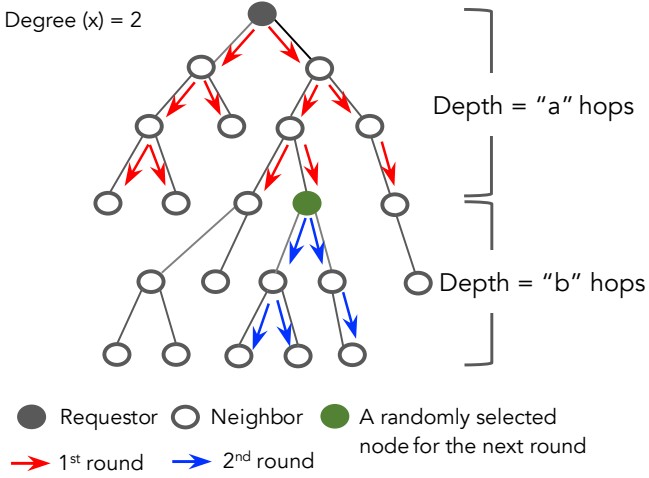

**Figure 2.** The policy of Random-X-Walker for propagating the query in multiple rounds.

### 4.2.2. Algorithm

In this approach, only the number of queries required by regions will be sent, which may entail many more rounds of sending and thus a longer time taken. For example, $10 - x$ regions would be the value for the current round of questioning, if we had already found $x$ regions. More precisely, in round $i$, if $k_i < k$ regions have already been found where $F$ evaluates to TRUE, we would query a further $k - k_i$ regions in which the crowd had not previously been queried. It can be seen that this solution never asks more questions than those as required, and hence, minimizes cost. However, at the same time, it provides a means to finish in fewer rounds than flooding. This solution is in selectNeighbors $(S, k, D, x, R \setminus O)$ shown in Algorithm 6. In contrast to Algorithm 2, in each round Algorithm 6 will randomly select only some neighbors (the number of nearby peers defined by the $X$ value) to query (calling it random spatial crowdsourcing) and limit queries using hops distance. The algorithm proposed later in the article uses a multi-criteria neighbor selection method to select regions to query.

### 4.2.3. Analysis

To analyze spatial crowdsourcing via this solution, the total cost for the best case is $k \cdot \phi$ (where $\phi$ refers to the paid rate for an answered query) with the least number of rounds being 1. Meanwhile,

in the worst case, the total cost is $|R| \cdot \phi$ with the largest number of rounds $|R|$. Let us consider the average case. If we assume that the probability of each traverse node for responding and the probability of forwarding the query are $\alpha$ and $\beta$ respectively, then we can describe the typical case in the algorithm via a success factor as $\sigma$. In random spatial crowdsourcing, we would have $\sigma = \alpha \cdot \beta$ such that $0 < \sigma \leq 1$, where $\alpha$, as given earlier, is the proportion of regions in $R$ where $F$ evaluates to TRUE and $\beta$ denotes the fraction of regions in $R$ where nodes who received the query are willing to forward it to others.

To drive the cost efficiency, we have to consider the appropriate number of queries propagated among regions. Note that the type of network in our case is a directed $n$-ary tree network, which is a rooted tree in which each node has exactly one path and no more than $n$ children. In our case, the degree is defined as the number of neighboring nodes who can receive a query from the requestor/peer within a communication range (e.g., Wi-Fi Direct, LTE Direct). With regard to query propagation, this type of network allows many queries to be simultaneously distributed to neighboring nodes at the same time. Therefore, the upper boundary on the number of regions ($c$) receiving the query is provided by Equation (2), where we assume that the network is a complete $n$-ary tree which has degree $x$ and a hop limit equalling $HL$ of the tree.

$$c = x \left( \frac{x^{HL} - 1}{x - 1} \right), \ where \ c \leq k \tag{2}$$

The Equation (2) can be derived as follows. At level 0 there is $x^0 = 1$ node. The next level has $x^1$ nodes, and so on, with $x^n$ nodes at level $n$. Hence, the total number of nodes will be $\sum_{i=0}^{HL} x^i = \frac{x^{HL+1}-1}{x-1}$. However, the root node (source) has to be subtracted since it cannot answer itself. Therefore, $M = \frac{x^{HL+1}-1}{x-1} - 1 = \frac{x(x^{HL}-1)}{x-1}$. From the Equation (2), if the requestor initializes the number of regions to find as $c$, the hop distance can be derived and given by Equation (3).

$$HL = \left\lceil \frac{\ln \left( \frac{c(x-1)}{x} + 1 \right)}{\ln (x)} \right\rceil \tag{3}$$

In the Equation (3), $x$ refers to the degree of a node assigned by the requestor and $c$ is defined as the number of unobserved regions, which can be calculated from $c = k - |D|$, as shown in Algorithm 6. To reduce costs and increase efficiency, $c$ is no more than $k$ (the number of regions required).

As mentioned, if the total number of regions answered positively has not reached the number of regions required, this approach lets the requestor query for multiple rounds. The algorithm used $k$ queries in round 1, $k - \lceil k.\sigma \rceil$ queries in round 2, given the expected $\lceil k.\sigma \rceil$ successes from round 1. Then there are $(k - \lceil k \cdot \sigma \rceil) - \lceil (k - \lceil k \cdot \sigma \rceil) \cdot \sigma \rceil$ queries in round 3; given $\lceil (k - \lceil k \cdot \sigma \rceil) \cdot \sigma \rceil$ successes from round 2, and so on. By doing this, we can find the total number of queries in this scheme. If $Q(n)$ denotes the total number of queries used to find n positive regions, then, $Q$ is determined by:

$$Q(0) = 0 \tag{4}$$

$$Q(n) = n + Q(n - \lceil n \cdot \sigma \rceil) \tag{5}$$

The reason for this is that, with a success rate of $\sigma$, to find $n$ positive regions we first use $n$ queries to find $\lceil n \cdot \sigma \rceil$ successes, and then to find the remaining $n - \lceil n \cdot \sigma \rceil$ positive regions, we use $Q(n - \lceil n \cdot \sigma \rceil)$ queries.

### 4.3. Random-X-Walker with Redundancy

#### 4.3.1. Idea

From the method above, the number of queries sent to peers in each round has to be equal to the remaining number of regions to be found in each round. This method is the most effective for cost reduction. However, it may decrease work efficiency since many rounds of queries may need to be sent in order to reach the expected number of regions in cases where there are less than 100% positives. To maintain a balance between cost and work efficiency, we introduced a redundant questioning scheme by having more queries sent to peers in each round.

#### 4.3.2. Algorithm

In this method, the value for $\gamma$ has been set. It refers to a multiplier applied to the number of expected regions; $\gamma$ is between $1 \leq \gamma \leq \frac{1}{\sigma}$, while $\sigma$ is the success factor ($\sigma = \alpha \cdot \beta$, as explained above). That is, we can obtain $k$ regions faster by having $\gamma \cdot (k - k_i)$ queries in the next round $i + 1$, where $k_i$ is the number of regions found to be *TRUE* so far, up to and including round $i$. In round $i + 1$, by asking more queries, the number of successes is then $\lceil (number\ of\ queries) \cdot \sigma \rceil = \lceil (\gamma \cdot (k - k_i)) \cdot \sigma \rceil$. This slight variation to the solution above is given by the definition of selectNeighbors($S, k, D, x, R \setminus O$) in Algorithm 7.

---

**Algorithm 7:** selectNeighbors($S, k, D, x, R \setminus O$)

---

**Input** : $S$: set of source nodes = $\{s_i, ..., s_n\}$, $k$: number of positive regions to find,
$D$: set of found regions, $R$: set of regions, $O$: set of observed regions,
$n_i$: set of selected unobserved neighbors of $s_i$, $x$: number of degrees

**Output:** $N$: set of selected unobserved neighbors to be queried

    $N := \varnothing$;
    $c := \lceil \gamma.(k - |D|) \rceil$;
    **for** each $s_i$
        $n_i :=$ randomly select $x$ neighbors within the communication range of $s_i$ from
        $R \setminus O$, where $|n_i| \leqslant c$;    ($n_i := c$ neighbors, when $x > c$)
        $N := N \cup n_i$;
  **return** $N$;

---

#### 4.3.3. Analysis

We call this Random-X-Walker with Redundancy (RXW-G), when $\gamma > 1$, and Random-X-Walker with no redundancy (RXW) when $\gamma = 1$ (the algorithm given earlier). In general, if $Q'_y(k)$ denotes the total number of queries used to find $k$ positive regions using this algorithm. Then, $Q'_y$ is determined by:

$$Q'_y(0) = 0 \tag{6}$$

$$Q'_y(n) = \lceil \gamma \cdot n \rceil + Q'_y(n - \lceil \lceil \gamma \cdot n \rceil \cdot \sigma \rceil) \tag{7}$$

To find $n$ positive regions, RXW-G starts with $\lceil \gamma \cdot n \rceil$ queries, finding $\lceil \lceil \gamma \cdot n \rceil \cdot \sigma \rceil$ positive regions, and then to find the remaining $(n - \lceil \lceil \gamma \cdot n \rceil \cdot \sigma \rceil)$ regions, it uses $Q'_y(n - \lceil \lceil \gamma \cdot n \rceil \cdot \sigma \rceil)$.

### 4.4. Multi-Criteria Neighbor Selection

#### 4.4.1. Idea

For the RXW and RXW-G approaches, the results are quickly returned to the source nodes; by using a random technique, the query message is forwarded to the source's neighbors. However, it appears that this approach cannot ensure cost reduction and work effectiveness due to the

inappropriateness of the randomly-found nodes when queries are distributed to regions; for example, nodes might refuse to forward tasks to their neighbors, causing inadequacies in the expected regions or even repeated calls for further queries.

There are some studies in [46–48] using historical/statistic data, such as workers' historical task completion performance and historical movement, to select participants for crowdsourcing systems. The historical and statistical data help in selecting the right workers and enhancing the task quality [48]. To solve the above problems, choosing "good" neighbors by keeping track of simple statistics on neighbors (e.g., the number of positive results returned through the neighbors and the number of queries forwarded to others) has been employed. These statistics-based data are collected locally in the workers' devices, and we assume that the data are accessible by the system. These data are useful in determining which nodes are most appropriate for query distribution to ensure the efficient attainment of the expected number of returns, reduce the number of task distribution rounds, and promote network power saving. The criteria for selecting candidate neighbors in the P2P spatial crowdsourcing system are as follows:

- the number of positive results returned previously
- the number of query messages forwarded to others previously
- the number of neighbor nodes
- the battery power remaining

We propose using the Multi-Criteria Neighbor Selection (MCNS) technique, which takes into account trustworthiness among regions in P2P spatial crowdsourcing. Each source node computes the value of their neighbors, which is then used for further query forwarding. In this approach, the value weights are assigned to different criteria. These metrics have different weights depending on the requestor's policy or performance requirements. For example, we may give higher weights to some metrics that are more sensitive to the performance in a P2P spatial crowdsourcing system. For $t$ weights, we have $\sum_{i=1}^{t} w_i = 1$, where $w_i$ is the weight assigned to metric i ($1 \leq i \leq t$). After a node has obtained enough information and created its neighbor list, the MCNS calculates the score for node $v$ using the Equation (8):

$$W_v = w_1 R_v + w_2 F_v + w_3 D_v + w_4 P_v \tag{8}$$

where $w_1, w_2, w_3$ and $w_4$ are the criteria weighs for the corresponding spatial crowdsourcing system parameters, $R_v$ is the ratio of positive answers in the node $v$; $F_v$ is the ratio of forwarded queries; $D_v$ is the ratio of the number of neighbors of a node $v$; and $P_v$ is the ratio of the remaining power of node $v$. Using this equation, neighbors with a higher $W_v$ are targeted for further query forwarding. The expectation is that there will be an increase in rates of successful query returns with corresponding reductions in payment costs and power consumption.

### 4.4.2. Algorithm and Analysis

The only difference between this approach and Random-X-Walker is in the node selection, which is described as follows:

1. Find the neighbors of each node v (i.e., nodes within its transmission range).
2. Compute a ratio using the number of positive answers derived from each of node $v$ neighbors, using the formula: $R_v = \dfrac{r_i}{\sum_{i=1}^{n} r_i}$, where $r_i$ is the total number of positive results previously returned from node $i$.
3. For every node, compute a ratio using the number of forwarded queries of node $v$, using the formula: $F_v = \dfrac{f_i}{\sum_{i=1}^{n} f_i}$, where $f_i$ is the total number of forwarded queries of node $i$.
4. Compute a ratio using the number of each of node $v$'s neighbors. This gives a relative measure of node degree and is denoted by $D_v$, as $D_v = \dfrac{d_i}{\sum_{i=1}^{n} d_i}$, where $d_i$ is the number of neighbors of node $i$.

5.  Then, compute the relative remaining power of node $v$, using the formula: $P_v = \frac{p_i}{\sum_{i=1}^{n} p_i}$, where $p_i$ is the remaining power of node $i$.

6.  Calculate the score of node $v$, using the formula:

$$W_v = w_1 R_v + w_2 F_v + w_3 D_v + w_4 P_v$$

where $w_1, w_2, w_3$ and $w_4$ are the weighing criteria for the corresponding spatial crowdsourcing and the sum of weights $w_1, w_2, w_3$ and $w_4$ is equal to 1.

7.  Choose the node with the highest $W_v$. A query will be then sent to that node.

8.  Repeat steps 2 to 7 for the remaining nodes not yet selected for forwarding any queries.

For task propagation, we have used Algorithm 3 with a slight variation to the solution above given by selectNeighbors$(S, k, D, x, R \setminus O)$, as shown in Algorithm 8. In Algorithm 8, the list *LN* is introduced to collect the neighbors of each of the source nodes except the ones already observed. Each neighbor has its score calculated using the above method in the function calculateScore$(l_j)$. Then these neighbors of Si are put in order starting from the highest score to the lowest. The $x$ highest scoring neighbors are selected for queries to be sent to them.

---

**Algorithm 8:** selectNeighbors$(S, k, D, x, R \setminus O)$

---

**Input** : $S$: set of source nodes = $\{s_i, ..., s_n\}$, $k$: number of positive regions to find,
$D$: set of found regions, $R$: set of regions, $O$: set of observed regions,
$n_i$: set of selected unobserved neighbors of $s_i$, $x$: node degree,
$LN$: a list of neighbors of $s_i$, $l_j..., l_n$: numbers of the list LN

**Output**: $N$: set of selected unobserved neighbors to be queried

    $N := \varnothing$;
    $c := k - |D|$;
    **for** each $s_i$
        $LN := \varnothing$;
        $LN :=$ select all neighbors within the communication range of $s_i$ from $R \setminus O$;
        **for** each $l_j$ of $LN$
            $l_j.score :=$ calculateScore$(l_j)$;
        **end**
        $LN :=$ sort $LN$ in descending order of their scores;
        $n_i :=$ select top $x$ neighbors from $LN$, where $|n_i| \leqslant c$;
            ($n_i := c$ neighbors, when $x > c$)
        $N := N \cup n_i$;
    **end**
    **return** $N$;

---

*4.5. Multi-Criteria Neighbor Selection with Redundancy*

4.5.1. Idea

We also propose a redundant questioning technique allowing more queries to be sent to peers in each round in MCNS called MCNS-G. In order to reach the expected number of regions as fast as they could, the value of $\gamma$ has been set: $\gamma$ is between $1 \leq \gamma \leq \frac{1}{\sigma}$, while $\sigma$ is the success factor ($\sigma = \alpha \cdot \beta$, as explained in Section 4.5.2). We can obtain $k$ regions faster by having $\gamma \cdot (k - k_i)$ queries in the next round $i + 1$, where $k_i$ is the number of regions found to be *TRUE* so far, up to and including round $i$.

### 4.5.2. Algorithm and Analysis

In round $i + 1$, by asking more queries, the number of successes is then $\lceil (number\ of\ queries) \cdot \sigma \rceil = \lceil (\gamma \cdot (k - k_i)) \cdot \sigma \rceil$. This slight variation to the solution above is given by the definition of selectNeighbors$(S, k, D, x, R \setminus O)$ in Algorithm 9.

---

**Algorithm 9:** selectNeighbors$(S, k, D, x, R \setminus O)$

---

**Input** : $S$: set of source nodes = $\{s_i, ..., s_n\}$, $k$: number of positive regions to find,
$\quad\quad\quad\quad$ $D$: set of found regions, $R$: set of regions, $O$: set of observed regions,
$\quad\quad\quad\quad$ $n_i$: set of selected unobserved neighbors of $s_i$, $x$: node degree,
$\quad\quad\quad\quad$ $LN$: a list of neighbors of $s_i$, $l_j...,l_n$: numbers of the list LN
**Output:** $N$: set of selected unobserved neighbors to be queried

$\quad\quad\quad$ $N := \varnothing$;
$\quad\quad\quad$ $c := \lceil \gamma.(k - |D|\rceil$;
$\quad\quad\quad$ **for** each $s_i$
$\quad\quad\quad\quad\quad$ $LN := \varnothing$;
$\quad\quad\quad\quad\quad$ $LN :=$ select all neighbors within the communication range of $s_i$ from $R \setminus O$;
$\quad\quad\quad\quad\quad$ **for** each $l_j$ of $LN$
$\quad\quad\quad\quad\quad\quad\quad$ $l_j.score :=$ calculateScore$(l_j)$;
$\quad\quad\quad\quad\quad$ **end**
$\quad\quad\quad\quad\quad$ $LN :=$ sort $LN$ in descending order of their scores;
$\quad\quad\quad\quad\quad$ $n_i :=$ select top $x$ neighbors from $LN$, where $|n_i| \leqslant c$;
$\quad\quad\quad\quad\quad\quad\quad$ ($n_i := c$ neighbors, when $x > c$)
$\quad\quad\quad\quad\quad$ $N := N \cup n_i$;
$\quad\quad\quad$ **end**
$\quad\quad$ **return** $N$;

---

## 5. Performance Evaluation

We conducted extensive simulation-based testing to study the distribution of crowd tasks in spatial crowdsourcing and conducted basic scalability experiments. We created a mobile crowdsourcing simulation model based on the basic operations of the *LogicCrowd* application [36,49,50] implemented on actual devices. We have extended the work from [14] for exploring the spatial crowdsourcing paradigm. Netlogo (https://ccl.northwestern.edu/netlogo/ [Last accessed 25 June 2020]) has been used for simulating two different aspects of crowd spatial networks in natural and social phenomena experiments: crowd task propagation and energy consumption. The purpose of the simulation presented in this section is two-fold: (1) to analyze the relationship between the parameters of spatial crowdsourcing and the number of responses, and (2) to study the cost and energy consumption characteristics of P2P spatial crowdsourcing, especially in a large scale ad-hoc network.

### 5.1. Simulation with Netlogo and Scenarios

Figure 3 shows the simulator interface for our testbed P2P spatial crowdsourcing network. Each node is associated with a region and they are linked with nearby neighbor nodes within communication distance, and then connected to the larger network. In the model, we defined the parameters related to our proposed idea, including time periods, task sizes, the number of neighbor nodes for each node, the probability of a peer responding, the probability of a peer forwarding, and the battery level remaining for each peer. Values were assigned for the parameter in the P2P spatial crowdsourcing model in order to study their impact on the number of responding regions and on the battery usage within the whole network across different simulation executions.

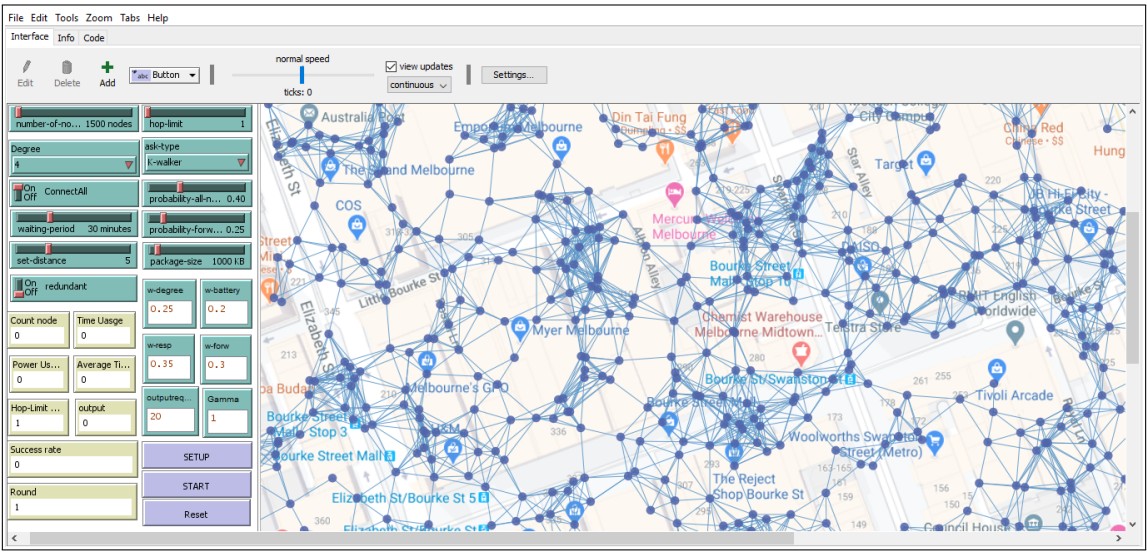

**Figure 3.** Simulation interface for the testbed peer-to-peer (P2P) spatial crowdsourcing network—note the geographical spread of the network in the background map.

All of the details about the parameter settings can be found in Table 1. We fixed the area which the network spans at $300 \times 300$ square meters (the ratio was 1 m = 1 patch in Netlogo). By creating nodes randomly positioned within this area, we generated 30,000 nodes. These 30,000 nodes were connected to each other with communication distances of 10 m (this is well within current Bluetooth ranges and also well within Wi-Fi Direct and vehicle-to-vehicle Dedicated Short Range Communications (DSRC) ranges). When any nodes were within a communication range, they were then assumed to be connected to the specified degree setting. Note that each node occupied a region so that a response from a region was a response from a node about that region.

**Table 1.** The parameters of scenarios.

| Parameters | Values for Scenarios |
|---|---|
| Area size | $300 \times 300$ m$^2$ |
| Communication Distances | 10 m |
| Number of nodes | 30,000 nodes |
| Probability of peers responding ($\alpha$) | 0.5 |
| Probability of peers forwarding ($\beta$) | 0.5 |
| Number of time ($\gamma$) | 1–4 |
| Waiting period ($wp$) | 5–25 min |
| Packet size ($pk$) | 100–2000 MB |
| Degree of node ($d$) | 1–10 |

*5.2. Comparing Approaches*

In this section, we present the results from our experiments on five approaches including (1) flooding (baseline), (2) Random-X-Walker (RXW), (3) Random-X-Walker with $\gamma$ (RXW-G), (4) Multi-Criteria Neighbor Selection (MCNS), and (5) Multi-Criteria Neighbor Selection with $\gamma$ (MCNS-G) that we proposed using various metrics: peer/region responses, energy consumption, cost, and time usage.

5.2.1. Evaluation Results

Figure 4 shows the comparison of the five different approaches. Figure 4a presents the number of regions responding (or found regions) for waiting periods from 5 min up to 25 min. It is apparent

that the longer the waiting period, the higher the number of responding regions, which we expected. Moreover, the number of regions responding using the flooding approach was the highest, whereas the lowest number of responses were from the RXW scheme, on average. Note that at a waiting period of between 5 to 10 min for all proposed methods, the number of region responses increased greatly; for instance, it increased from 5112 to 15,000 regions in the flooding approach and from 36 to 140 regions in the RXW technique, while the number of found regions only grew slightly increasing the waiting period to 15 to 25 min. Indeed, the flooding approach was, in this sense, more time sensitive compared to the other (more controlled dissemination) approaches.

One of the factors that impacts the number of region responses is the packet sizes. In Figure 4b, we can see that four methods including RXW, RXW-G, MCNS and MCNS-G shared the same trend; that is, there was a slight decrease in the number of found regions when the packet sizes of the nodes rose from 100 KB to 2000 KB. For example, finding the regions via RXW scheme at a waiting period of 25 min, nodes degree of 3 and $\gamma$ value of 4, but with varying packet sizes of 100, 500, 1000, 1500 and 2000 KB, the number of returned answers was 63, 42, 55, 24, and 30 regions, respectively. It should be noted that MCNS-G method obtained the highest number of responses, followed by MCNS, RXW-G, and RXW respectively. The number of returned answers in MCNS-G were 2362, 1735, 1067, 887, and 650 regions. This is in contrast to the results of the flooding technique when the packet size increased from 100 KB to 2000 KB. It clearly shows that although this method received the highest number of found regions among all of the approaches, it remained at 15,000 regions for all five packet sizes. This also indicates the sensitivity to the size of query messages and answers and how they approached the scale with increasing message sizes (up to 2000 KB).

In our study, we also measured the power consumption of (assumed) mobile (peer) applications running in each node via changes in battery levels. Although battery level changes are coarse-grained, they can be easily collected through a user-level application on mobile devices. To estimate the energy used in this study, we exploited the energy consumption model from our previous work in [36] performed on real devices. We predicted that there is a power consumption rate for the task-downloading process of 0.017% (of battery life) and a power usage rate for the task-forwarding process of 0.024% when carrying a 100 KB packet. When measuring energy consumption, we based it on the model as given, but there are uncertainties here as more likely, on a real-device, there could be factors that affect the energy consumption which the simplified model does not account for, but our simplified model serves as a common basis, useful for comparisons of the different algorithms.

Figure 4c shows the total percentages of changes in battery levels used from the entire network with varying waiting periods (5–25 min), comparing the five approaches with a node degree of three and data packet size of 100 KB. It clearly shows that the power consumption for the entire network with the flooding method was significantly higher than the others, followed by MCNS-G, MCNS, RXW-G and RXW respectively. Note that the power consumption of the other four methods shared the same trend; they grew slightly in a simple linear form as the waiting period moved from 5 up to 25 min. For example, the energy consumed by RXW, RXW-G, MCNS and MCNS-G methods with a node degree of three, packet sizes 100 KB, $\gamma$ value of 4 at 10 min were about 29.42%, 21.17%, 207.3% and 271.78% respectively (note that we are estimating the total energy consumed by nodes across the whole network, and so 271.78% means energy equivalent to approximately 2.7 fully charged devices). It is apparent that with a longer waiting period, more energy is consumed. However, the total energy used in the flooding method after 10 min remained steady despite an increase in the waiting period from 10 to 25 min (consumed energy of around 2370.05%). The reason is that the flooding scheme could distribute the tasks across all of the nodes in the experimental area (300 $\times$ 300 m$^2$ with 30,000 nodes, probability of peers forwarding and responding = 0.5) within 10 min. Hence, when we set the waiting period of more than 10 min, the total energy usage of the network did not change.

Figure 4d reports the total energy consumption when finding regions with the various packet sizes (from 100 to 2000 KB). There is a dramatic difference in battery consumption between flooding and the other four approaches. Note that the total energy consumption of the peers in the network

using flooding was considerably higher than the power consumption of the other four approaches. However, all five methods shared the same trend, making it readily apparent that with larger packet sizes, more energy was consumed. For instance, using flooding, the energy spent was about 2378%, 9215%, 18,080%, 29,000%, and 36,260% at packet sizes from 100 to 2000 KB, respectively, with a waiting period of 25 min, a degree of 3, and $\gamma$ value of 4. The MCNS-G method followed in second, with a total network battery usage, under the same conditions, of about 448.42%, 1401.02%, 1663.78%, 1988.55%, and 2046.25% at packet sizes from 100 to 2000 KB. Meanwhile, the energy spent by using the MCNS approach was about 239.4%, 1005.48%, 1287.4, 1334.98, and 1404.94% for packet sizes of 100, 500, 1000, 1500 and 2000 KB with a waiting period of 25 min, a degree of 3, and $\gamma$ value of 4. Finally, the RXW approach expended the least energy, with around 18.7% for a packet size of 100 KB and 302.56% for a packet size of 2 MB, under the same conditions.

In the experiments, we observed the relationship between the total time of usage and the number of regions required by the requestor for each of the approaches. In Figure 4e, the results show that all five methods have the same trend with a simple linear increase. Additionally, the RXW method took the longest time to reach the number of regions required since this method requires several rounds of region finding regardless of limiting the degrees of the nodes and the number of hops—randomization caused some nearer regions to be missed or peers who forwarded queries to be missed. Conversely, flooding took the shortest due to only needing one round of finding regions without any conditions. Moreover, we conducted the experiment to measure the cost of data transmission.

Figure 4f shows the comparison between the cost of data transmission and the number of regions required among the five approaches. We estimated the cost by applying the data transfer rate of a typical 4G network; that is, a data transmission of 100 KB was valued at $0.001. We found that MCNS-G spent the least money, followed by MCNS, RXW-G, flooding, and RXW in ascending order. The cost of using the MCNS-G approach at a specified number of regions (10, 30, 50, 80 or 100) with a packet size of 100 KB, and a 25 min-waiting period, with degree of 3, and $\gamma$ value of 4 were 0.02, 0.05, 0.09, 0.16, 0.22 dollars respectively. The reason is that appropriate neighbors were chosen by MCNS-G for query propagation; hence, there was a greater probability of these nodes to respond and forward the tasks in comparison to other approaches. In contrast, the cost of data transmission for RXW was the highest. When the requested number regions was increased from 10, to 30, 50, 80 or 100 regions, under the same conditions as above, the cost of data transmission was 0.04, 0.12, 0.18, 0.30 and 0.36 dollars respectively. Due to RXW using a random technique to select neighbors, the randomization was more likely to find nodes which would not respond to or forward the query to their neighbors, causing repeated rounds in an attempt to find the sufficient number of regions and, thereby, raising the total expenses.

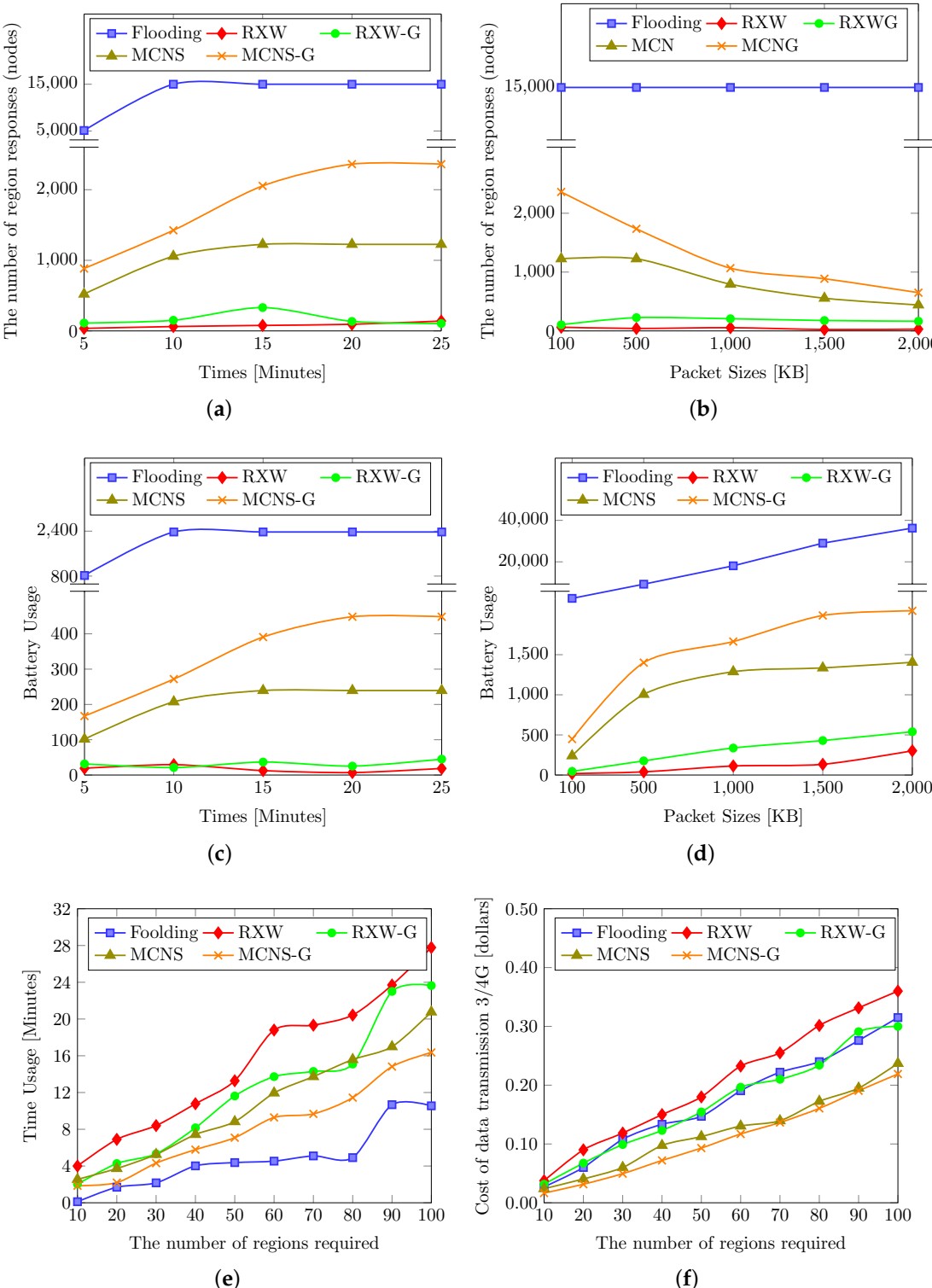

**Figure 4.** The comparison among five different approaches including (1) flooding, (2) Random-X-Walker (RXW), (3) Random-X-Walker with $\gamma$ (RXW-G), (4) Multi-criteria Neighbor Selection (MCNS) and (5) Multi-criteria Neighbor Selection with $\gamma$ (MCNS-G) in varying metrics including peer/region responses, energy consumption, query costs and time usage. (**a**) The relationship between the number

of region responses and waiting period with $d = 3$, $\gamma = 4$, $pk = 100$ KB; (**b**) The relationship between the number of region responses and packet sizes with $d = 3$, $\gamma = 4$, $wp = 25$ min; (**c**) The relationship between battery usage and waiting period with $d = 3$, $\gamma = 4$, $pk = 100$ KB; (**d**) The relationship between battery usage and packet sizes with $d = 3$, $\gamma = 4$, $wp = 25$ min; (**e**) The relationship between time usage for finding the required number of regions and the number of regions required with $d = 3$, $\gamma = 4$, $pk = 100$ KB; (**f**) The relationship between cost of data transmission and the number of regions required with $d = 3$, $\gamma = 4$, $pk = 100$ KB.

### 5.2.2. Discussion

The detailed results described in the previous section are summarized in Table 2. Table 2 presents a comparison of the five approaches. In this table, all of the methods are assessed relative to one another using the same metrics. High, moderate, and low rankings in this table are represented comparatively.

In Table 2, we find that there is no single approach which is optimal for all metrics. Each of them has their own limitations. This also implies that a system supporting P2P crowdsourced querying will need to support multiple strategies depending on the user's priorities at the time.

**Table 2.** Comparing the various methods.

| Metrics | Flooding | RXW | RXW-G | MCNS | MCNS-G |
|---|---|---|---|---|---|
| Number of region responses * | High | Low | Low | Moderate | Moderate |
| Battery usage * | High | Low | Low | Moderate | Moderate |
| Time usage ** | Low | High | Moderate | Moderate | Moderate |
| Cost (number of queried regions) ** | Moderate | High | Moderate | Moderate | Low |

* (for a given waiting period); ** (to find the required number of positive regions).

This result indicates there will be trade-offs among the following factors: the number of regions required, total amount of battery usage, cost, and latency. With queries spreading to all of a node's neighbors, the flooding approach is able to find the desired number of positive regions within a relatively short period of time. This approach can thus be a good choice regardless of the cost-efficiency and power consumption factors. However, if the budget for the cost of queries is limited, MCNS-G seems to be a better choice. By choosing good neighbors from historical data and considering the network characteristics of the regions, MCNS-G can strike a balance among the number of region responses, battery consumption, and total time spent within a fixed budget. Comparing the results for MCNS-G and MCNS, we found the similar trends in the number of region responses, battery consumption, and time usage. However, MCNS is very likely to cost more than MCNS-G because the $\gamma$ value is not used in this approach. Hence, the queries need to be sent in multiple rounds in order to find the required number of regions.

In addition, the performance of RXW is viewed as the poorest with the smallest number of regions found while requiring a long time period to search its neighbors, thereby incurring higher transmission costs in spite of using relatively less battery power. The result can be explained by the impact of the randomized method used for finding neighbors. This method may limit the probability of finding neighbors active in task forwarding, thereby resulting in queries being repeatedly submitted. Furthermore, compared to RXW, the RXW-G approach was found to take less time and have a lower cost for query sending. It seems possible that this result is due to the potential of $\gamma$, which can distribute more queries in a round, leading to a higher probability of finding the required number of positive regions and reducing costs incurred.

### 5.3. Comparison between RXW and RXW-G

In this section, we compare the performance of RXW and RXW-G, using different values of $k$ (the number of regions to find) and different values of $\sigma$ (the probability/success factor) with varying

values of $\gamma$ (the multiplier for the number of the required regions). We also fix the area over which the network spans to $100 \times 100$ square meters with the total number of regions being 3000, a packet size of 100 KB with a waiting period of 25 min and a node degree of 3. Note that the probability of each traversed node for responding and forwarding the query are $\alpha$ and $\beta$, respectively; we also define $\sigma = \alpha \cdot \beta$. Below, we give the average number of queries, the average number of rounds and the average number of hops over 20 runs. Note that RXW is the case of RXW-G with $\gamma = 1$.

### 5.3.1. Evaluation Results

The results shown in Figure 5 compare RXW and RXW-G using a run with $k = 20$ and Figure 6 in a run with $k = 100$. In the experiments, we observed the relationship among the average number of the factors as follows: queries sent to nodes, rounds and hops where the horizontal axis is labelled with the different values of $\gamma$. These results are shown in Figure 5a–f are presented with varying probabilities for each traversed node that were set to 0.16, 0.19, 0.25, 0.32, 0.45 and 0.81, respectively. We make the following observations.

In all cases of $\sigma$ tested, it was seen that the smaller $\sigma$ was, the more queries were delivered, resulting in an increase in the average number of hops and average number of rounds. For example, over the 20 runs represented in Figure 5a with $k = 20$ and $\sigma = 0.16$, the average numbers of queries were 120.75, 119.55, 116.30, 112.45 and 110.45 queries, with an average of 19.2, 9.4, 4.4, 4.05 and 2.9 rounds for nodes distributed in 28.65, 19.60, 11.80, 10.70 and 9.15 hops when the values of $\gamma$ were 1 (RXW with no redundancy approach), 2.313, 3.625, 4.938 and 6.25 respectively. However, in Figure 5f with $k = 20$ and $\sigma = 0.81$, the average numbers of queries were 22.75, 24.0, 22.4, 22.9 and 22.7, with averages of 2.2, 2.75, 1.0, 1.05 and 1.25 rounds for nodes distributed in 4.2, 4.85, 3.0, 3.15 and 3.35 hops when the values of $\gamma$ were 1, 1.063, 1.125, 1.188, 1.25, respectively.

For all values of $\sigma$ tested in Figure 5a–f, we can see that the value of $\gamma$ does not seem to be significantly correlated with the number of queries sent to regions. Considering the graphs in Figure 5, the number of queries remained steady even when the value of $\gamma$ increased. For example, averaging over 20 runs with $\sigma = 0.19$ in Figure 5b, the resulting values of $\gamma$ were 1 (RXW), 2.065 3.13, 4.195 and 5.26. The average numbers of queries sent to nodes were 27.95, 27.45, 26.7, 27.9 and 27.6, respectively. Moreover, the average numbers of queries with $\sigma = 0.25$ in Figure 5c were 42.45, 43.60, 40.35, 41.2 and 40.5, where the values of $\gamma$ were 1 (RXW), 1.75, 2.5, 3.25 and 4, respectively. In contrast, both rounds and hops were negatively related to the value of $\gamma$. This means that the larger the number of $\gamma$ was, the fewer rounds and hops were required. For illustration, the values of $\gamma$ in Figure 5d were 1(RXW), 1.531, 2.063, 2.594 and 3.125, yielding the average numbers of rounds of 7.75, 5.5, 6.0, 5.25 and 3.7 for nodes distributed in 12.35, 10.85, 13.4, 12.0 and 9.15 hops, respectively. While the values of $\gamma$ in Figure 5e were 1 (RXW), 1.305, 1.61, 1.915 and 2.22, the average numbers of rounds taken to find the targeted regions were 5.3, 3.5, 2.95, 2.85 and 2.4 and there was a decline in the number of hops to 8.75, 6.5, 6.05, 6.15 and 5.45 hops, respectively.

Moreover, a significant change in the value of either $\alpha$ (the probability of positive regions) or $\beta$ (the probability of regions forwarding the queries to others) can affect the number of queries sent to each region in different ways. We found notable differences between $\alpha$ and $\beta$ in Figure 5a and Figure 5b, where the $k$ stood at 20. In Figure 5a, the value of $\sigma$ was 0.16 as derived from $\alpha = 0.2$ and $\beta = 0.8$. Whereas, that of $\sigma$ in Figure 5b was 0.19 as derived from $\alpha = 0.75$ and $\beta = 0.25$. The results in Figure 5a showed that there were a considerable number of queries sent to other regions, e.g., when the value of $\gamma$ is 1, over 120.75 queries had to be delivered to regions, and at the value of $\gamma = 3.626$, there were over 116.3 queries sent. Conversely, in Figure 5b, where the value of $\gamma$ was 1, only 27.95 queries had to be distributed to regions, and when the value of $\gamma$ was 3.13, only 26.7 queries were sent.

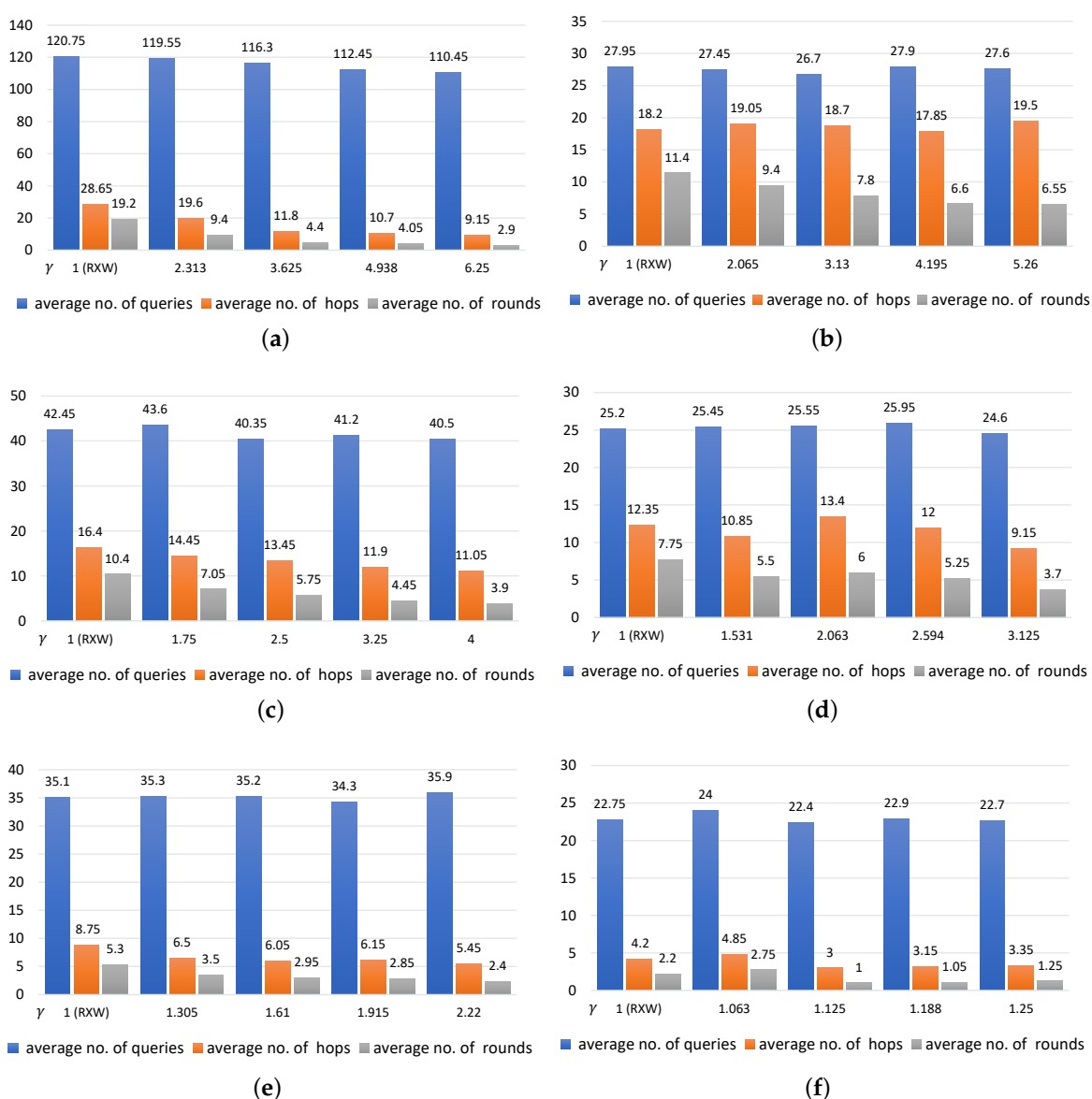

**Figure 5.** Comparing RXW and RXW-G: the number of queries sent to regions and number of hops and number of rounds averaged over 20 runs (vertical axis)for varying $\gamma$ (horizontal axis) for different values of $\sigma$ with $k = 20$. (**a**) $\sigma = 0.16$ ($\alpha = 0.20$, $\beta = 0.80$); (**b**) $\sigma = 0.19$ ($\alpha = 0.75$, $\beta = 0.25$); (**c**) $\sigma = 0.25$ ($\alpha = 0.50$, $\beta = 0.50$); (**d**) $\sigma = 0.32$ ($\alpha = 0.80$, $\beta = 0.40$); (**e**) $\sigma = 0.45$ ($\alpha = 0.60$, $\beta = 0.75$); (**f**) $\sigma = 0.81$ ($\alpha = 0.90$, $\beta = 0.90$).

The findings shown in Figure 5a can be explained by the different values of the probabilities. The probability for positive regions ($\alpha$) was set low and the probability for regions forwarding the queries ($\beta$) was set high. As a result, a smaller number of positive regions were returned to the requestor in each round, leading to a need for more query distributions in the following rounds. In contrast, when the value for $\alpha$ was set high and the value of $\beta$ set low for the test case as shown in Figure 5b, a large number of positive regions was returned to the requestor and fewer queries were sent to regions in the following rounds. Due to the lower number of regions forwarding the queries, the number of rounds and hops may increase when compared to those in the first example.

Moreover, we extended the experiment by increasing the number of regions required (i.e., $k$) to 100 regions. These findings are shown in Figure 6. The results from the experiment further reveals similar patterns with RXW-G over the same intervals for $\sigma$ with $k = 20$. For example, with over 20 runs

in Figure 6a with $k$ = 100 and $\sigma$ = 0.16, the average numbers of queries were 529.85, 531.25, 515.25, 516.3, and 524.25, with averages of 28.7, 13.9, 8.5, 7.6, and 6.0 rounds for nodes distributed in 65.45, 42.25, 29.7, 27.95, and 25.1 hops when the values of $\gamma$ were 1 (RXW with no redundant approach), 2.313, 3.625, 4.938, and 6.25, respectively. However, in Figure 6f with $k$ = 100 and $\sigma$ = 0.81, the average numbers of queries were 112.35, 113.3, 112.35, 112.55, and 111.65, with averages of 2.7, 2.4, 3.0, 2.25, and 2.45 rounds for nodes distributed in 6.95, 6.8, 7.95, 6.85, and 6.7 hops when the values of $\gamma$ were 1, 1.063, 1.125, 1.188, and 1.25, respectively.

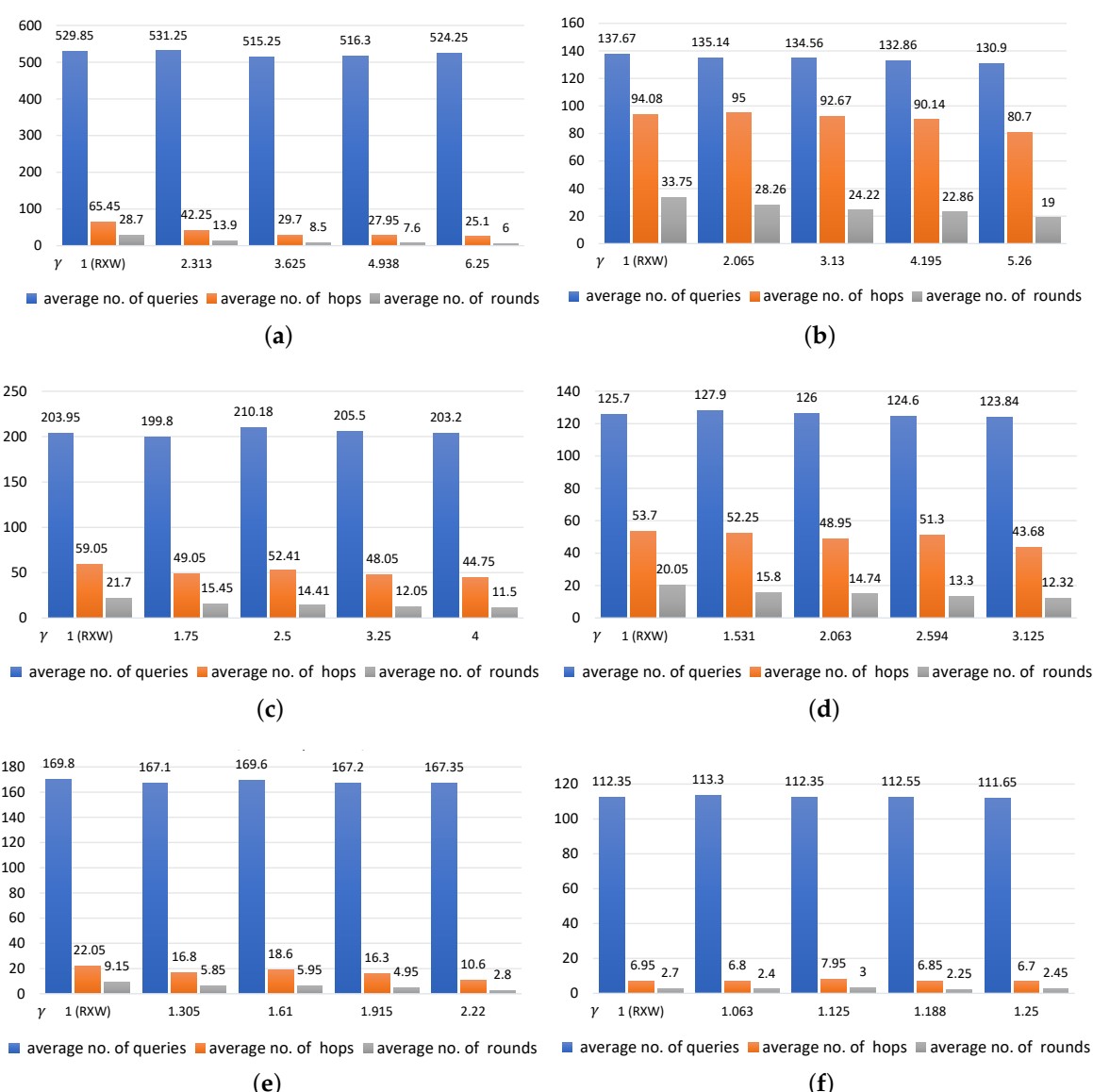

**Figure 6.** Comparing RXW and RXW-G: the number of queries sent to regions and number of hops and number of rounds averaged over 20 runs (vertical axis) for varying $\gamma$ (horizontal axis) for different values of $\sigma$ with $k$ = 100. (**a**) $\sigma = 0.16$ ($\alpha = 0.20$, $\beta = 0.80$); (**b**) $\sigma = 0.19$ ($\alpha = 0.75$, $\beta = 0.25$); (**c**) $\sigma = 0.25$ ($\alpha = 0.50$, $\beta = 0.50$); (**d**) $\sigma = 0.32$ ($\alpha = 0.80$, $\beta = 0.40$); (**e**) $\sigma = 0.45$ ($\alpha = 0.60$, $\beta = 0.75$); (**f**) $\sigma = 0.81$ ($\alpha = 0.90$, $\beta = 0.90$).

With $k$ = 100, it was clearly seen that the number of queries, rounds, and hops were affected in different ways when the values of $\alpha$ and $\beta$ were set differently. In Figure 6a with $\alpha$ = 0.2 and $\beta$ = 0.8, there were over 529.85 queries distributed on the network with an average of 28.7 rounds for regions

that were distributed over 65.45 hops where the $\gamma$ stood at 1. On the other hand, Figure 6b when using an $\alpha = 0.75$ and $\beta = 0.25$, and setting the value of $\gamma$ as 1, only 137.67 queries had to be distributed to regions, but the number of rounds and the number of hops rose to 33.75 and 94.08, respectively.

### 5.3.2. Discussion

From the detailed results described in the previous section, we can summarize the main points of the observations as follows. The results show that if we ask $1 \leq \gamma \leq \frac{1}{\sigma}$ more queries (the additional questions are the so-called redundant queries) than the minimum required in each round, we can significantly reduce the number of rounds and the number of hops in the RXW-G approach. Furthermore, because we reduce the number of rounds and the number of hops, we end up by only asking a small number of additional queries than we needed in total when compared to the no redundancy option. We see that this result also holds for a large range of scenarios. Moreover, the additional queries in each round lead to faster convergence towards the required number of positive regions (i.e., taking fewer rounds), although making more queries in the earlier rounds. Therefore, the faster convergence offsets the larger number of queries made in earlier rounds so that, overall, only a small percentage increase in the total number of queries is required, i.e., there is only a relatively small price to pay for faster results.

However, it must be noted that the gains are greater only when $\sigma$ is small and there is a large enough $\gamma$, and the value of $\gamma$ relies on some prior knowledge (or estimate) of $\sigma$, which might be difficult to obtain in practice. Using too small of a $\gamma$ does not result in much of an improvement in efficiency, but using too large of a $\gamma$ wastes queries. The resulting average of 1 round in Figure 5 and Figure 6 were all obtained with $\gamma$ having the value of approximately $\frac{1}{\sigma}$, as for each of the $k \cdot \frac{1}{\sigma}$, queries randomly chooses a region with $\frac{1}{\sigma}$ chance of being positive.

### 5.4. Comparison between MCNS and MCNS-G

In this section, we compare the performance of MCNS and MCNS-G, using different values of $k$ and different values of $\sigma$ with varying values of $\gamma$. The average number of queries, average number of rounds, and average number of hops are shown below. The experiment was preformed over 20 runs with the area size of the network spanning $100 \times 100$ square meters containing a total of 3000 regions, a packet size of 100 KB with a waiting period of 25 min, and a node degree of 3. Note that MCNS is the case of MCNS-G with $\gamma = 1$.

### 5.4.1. Evaluation Results

In a comparison of the results for MCNS and MCNS-G, which are shown in Figure 7 in a run with $k = 20$ and Figure 8 in a run with $k = 100$, we observed a relationship among the average values of the numbers of queries sent to nodes, rounds, and hops, where the horizontal axis is labelled with different values of $\gamma$. The results shown in these figures are presented with varying probabilities for each traversed node: 0.16, 0.19, 0.25, 0.32, 0.45, and 0.81. The results from the experiment further reveal similarities in the patterns of MCNS-G and RXW-G over the intervals. We make the following observations.

- The smaller $\sigma$ was, the more queries, hops, and rounds were needed, For example, with $\sigma = 0.16$, 0.32, 0.81 in Figure 7a,c,f and with $\gamma = 1$, increasing the values of $\sigma$ led to a slight drop in the number of queries (34, 25.6, and 21.2 queries, respectively). Under the same conditions, the average number of rounds taken to find the targeted regions were 3.4, 2.15, and 1.05 and there was a decline in the average number of hops to 5.4, 4.1, and 2.3, respectively.
- A likely explanation for why the value of $\gamma$ did not seem to be significantly correlated with the number of queried regions is that the number of queries remained steady even when the value of $\gamma$ increased. In contrast, both the average number of rounds and average number of hops were negatively related to the value of $\gamma$. This means that the larger the number of the $\gamma$ implied,

the lower the number of rounds and hops were. Note that this can be clearly seen when the $k$ values were large.

- The findings of this study suggest that a significant difference in either the value of $\alpha$ or of $\beta$ can affect the number of queries sent to each region in different ways. In cases where the $\alpha$ value was set low and the $\beta$ value was set high, only a small number of positive regions were returned to the requestor in each round, leading to the need for more query propagation in the following rounds. Conversely, if the value of $\alpha$ was fixed high and the value of $\beta$ was set low, a large number of positive regions were returned to the requestor and fewer queries were sent to regions in the rounds that followed. As a result, the numbers of rounds and hops may increase when compared to those in the previous case.

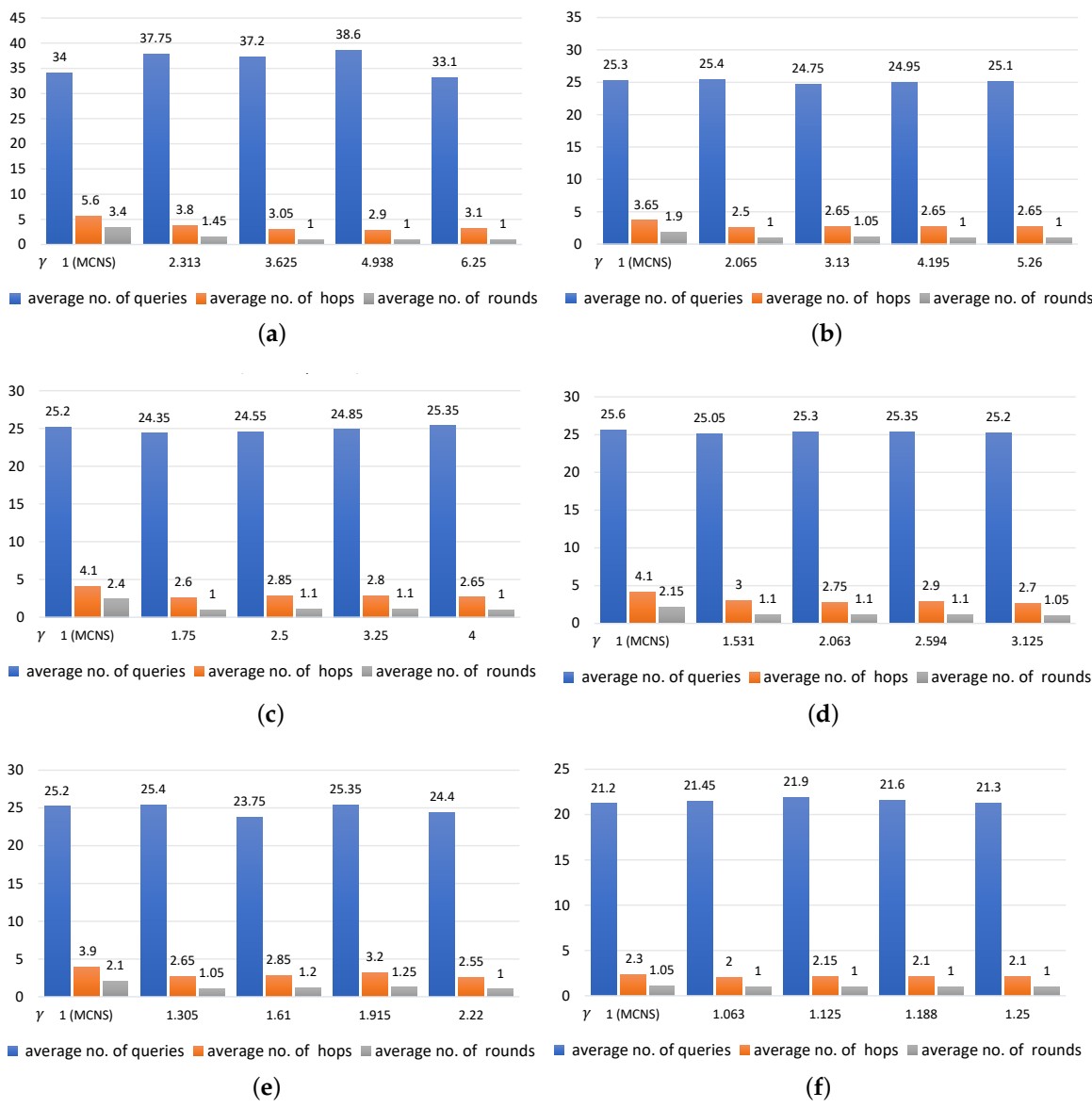

**Figure 7.** Comparing MCNS and MCNS-G: the number of queries sent to regions and number of hops and number of rounds averaged over 20 runs (vertical axis)for varying $\gamma$ (horizontal axis) for different values of $\sigma$ with $k = 20$. (**a**) $\sigma = 0.16$ ($\alpha = 0.20$, $\beta = 0.80$); (**b**) $\sigma = 0.19$ ($\alpha = 0.75$, $\beta = 0.25$); (**c**) $\sigma = 0.25$ ($\alpha = 0.50$, $\beta = 0.50$); (**d**) $\sigma = 0.32$ ($\alpha = 0.80$, $\beta = 0.40$); (**e**) $\sigma = 0.45$ ($\alpha = 0.60$, $\beta = 0.75$); (**f**) $\sigma = 0.81$ ($\alpha = 0.90$, $\beta = 0.90$).

However, the numbers of queries distributed to regions were found to be lower in MCNS-G than in RXW-G. Hence, the MCNS-G required lower power consumption and incurred lower expenses, possibly implying it was more effective.

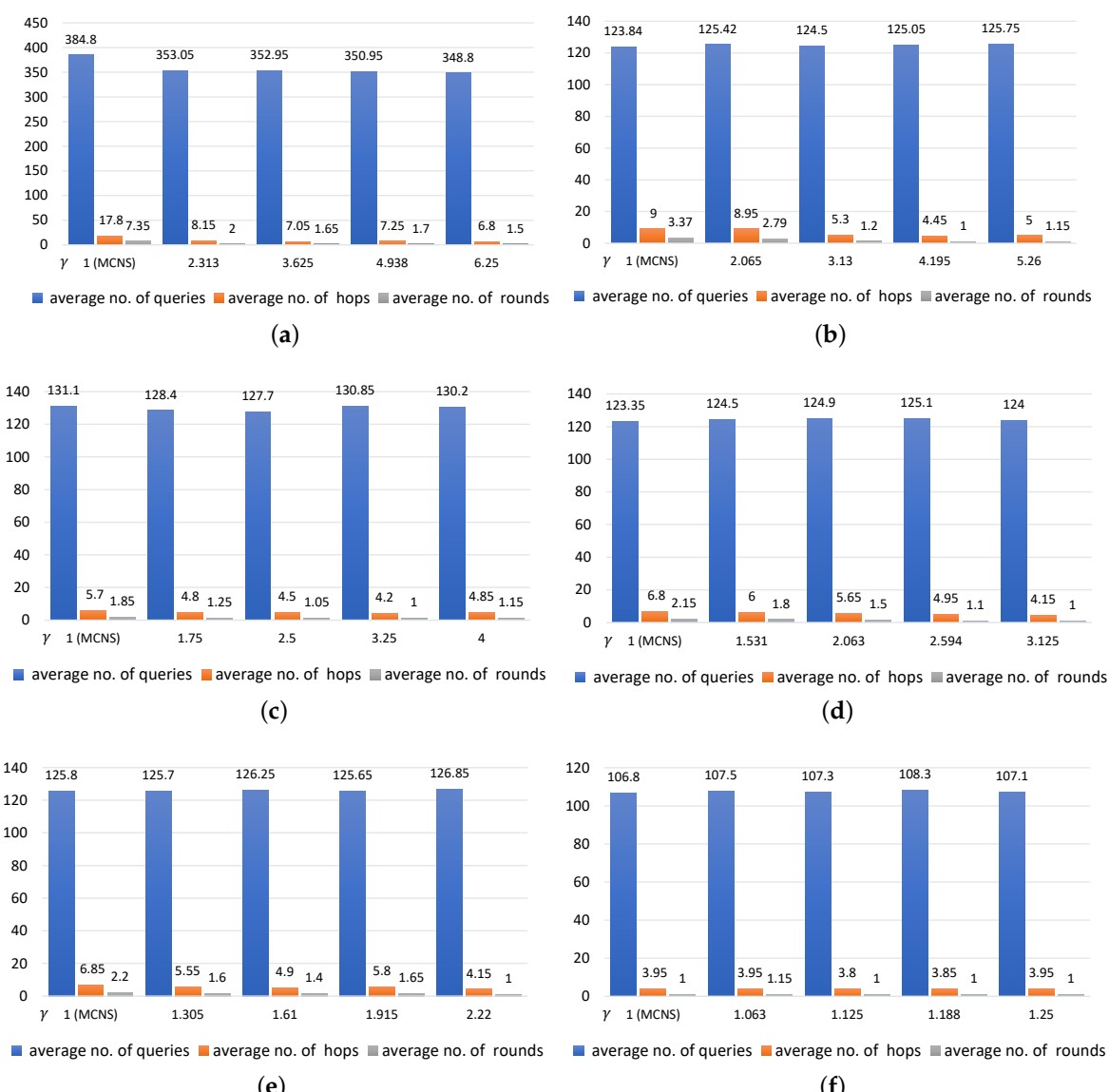

**Figure 8.** Comparing MCNS and MCNS-G: the number of queries sent to regions and number of hops and number of rounds averaged over 20 runs (vertical axis) for varying $\gamma$ (horizontal axis) for different values of $\sigma$ with $k = 100$. (**a**) $\sigma = 0.16$ ($\alpha = 0.20$, $\beta = 0.80$); (**b**) $\sigma = 0.19$ ($\alpha = 0.75$, $\beta = 0.25$); (**c**) $\sigma = 0.25$ ($\alpha = 0.50$, $\beta = 0.50$); (**d**) $\sigma = 0.32$ ($\alpha = 0.80$, $\beta = 0.40$); (**e**) $\sigma = 0.45$ ($\alpha = 0.60$, $\beta = 0.75$); (**f**) $\sigma = 0.81$ ($\alpha = 0.90$, $\beta = 0.90$).

## 5.4.2. Discussion

Based on the above results, by using the MCNS method, we can reduce the number of queries, hops, and rounds by 19.8%, 72.9%, and 75.1%, when compared to the RXW method. Meanwhile, similar results were found using the MCNS-G scheme, where there is a decrease in the number of queries, hops, and rounds of 21.1%, 73.9%, and 77.4% when compared to RXW-G. Moreover, the findings show that if we ask $1 \leq \gamma \leq \frac{1}{\sigma}$ more queries than the minimum required in each round, we can significantly reduce the number of rounds and the number of hops in the MCNS-G approach. It must be noted that

the gains are greater only when the $\sigma$ is small and the $\gamma$ is large enough. Additionally, using too small $\gamma$ does not result in much improvement in efficiency, but using too large $\gamma$ wastes queries.

## 6. Conclusions

In this paper, we proposed and investigated finding regions of interest from a set of regions in an area using iterative crowdsourcing in P2P networks, where queries are done over multiple rounds when each round has a carefully selected number of queries and peers to query. We have described five approaches incorporated into these crowdsourcing algorithms, to reduce cost (the number of queries required) and increase the efficiency (reducing the number of rounds of querying required) in using crowdsourcing to find regions of interest.

- Using flooding technique to propagate a query to all of a node's neighbors neighbors. During the propagation process, time-to-live (TTL) values are defined to limit the lifetime of queries so that the action of forwarding queries to others would eventually be stopped. This approach is able to find the desired number of positive regions within a relatively short period of time. This approach can thus be a good choice regardless of the cost-efficiency and power consumption factors.
- Using randomly forwarded queries over regions to X neighbors. This random walker mechanism executes until the TTL value expires or it receives *k* region responses. Moreover, we also proposed an efficient method using a proportionate number of redundant queries ($\gamma$ value) in each round in the expectation of failures to respond. The RXW method seems to have the poorest performance with the smallest number of regions, thereby resulting in queries being repeatedly submitted. In contrast, due to the potential of $\gamma$, the RXW-G approach can distribute more queries in a round, leading to a higher probability of finding the positive regions.
- Using historical or statistic data, such as workers' historical task completion performance and movement, to select participants for spatial crowdsourcing. Choosing "good" neighbors by keeping track of simple statistics on a node's neighbors (e.g., the number of positive results returned through the neighbors and the number of queries forwarded to others) has been deployed. In addition, in this technique, we also used a proportionate number of redundant queries ($\gamma$ value) in each round in the expectation of failures to respond or negative responses. By choosing good neighbors from historical data and considering the network characteristics of the regions, MCNS-G can strike a balance among the number of region responses, battery consumption, and total time spent within a fixed budget.

We demonstrated, via creating a mobile crowdsourcing simulation model, the results of our approaches and investigated their relative advantages. The MCNS approach leads to the improved performance over randomly choosing regions to query. Using a proportionate number of redundant queries in each round (MCNS-G) is able to significantly reduce the number of rounds and hops needed, thus improving efficiency in finding the required number of positive regions. This has implications on applications of crowdsource over peer-to-peer networks, which should take into account the multiple criteria for effective propagation of queries, including each node's responsiveness to queries, energy consumption, time taken and payment for answers.

For future work, we plan to implement the algorithms on a large set of participating smartphone devices to consider real-world test scenarios with actual transmissions and to address the issues of varying battery power and heterogeneous power consumption rates. We also plan to further examine practical applications of our approach, e.g., crowdsourcing to discover in real-time broadband networking coverage of a local area, to find car parking, to search for particular items within the vicinity, to discover where crowds and events are in a city, and to seek recommendations from locals. While we investigated a range of algorithms, there are further possibilities to explore, e.g., algorithms which take into account knowledge of intermediaries about who or which other device could possibly be the most helpful for the given query, and to forward accordingly—this means that different queries would be forwarded differently, instead of forwarding each query in the same way. In this work, we have

used the uniform probability distribution in our randomized selections, but there could be other distributions that could be used, and deeper analysis of other selection techniques can be explored.

**Author Contributions:** Conceptualization, S.W.L. and J.P.; Methodology, J.P.; Software, J.P.; Validation, J.P.; Formal Analysis, S.W.L. and J.P.; Data Curation, J.P.; Writing—Original Draft Preparation, J.P.; Writing-Review and Editing, S.W.L. and J.P.; Visualization, J.P. All authors have read and agree to the published version of the manuscript.

**Funding:** This research received no external funding.

**Conflicts of Interest:** The authors declare no conflict of interest.

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
