# Peer review of "Iterative Spatial Crowdsourcing in Peer-to-Peer Opportunistic Networks"

_electronics, doi:10.3390/electronics9071085_

Round 1

Reviewer 1 Report

In this paper, the authors proposed and investigated finding regions of interest from a set of regions in an area using iterative crowdsourcing in P2P networks, where queries are done over multiple rounds when each round has a carefully selected number of queries and peers to query. The authors have described five approaches incorporated into these crowdsourcing algorithms. Overall the motivation is strong, the overall quality and the problem addressed is interesting.

It would be better if the authors could improve the writing particularly English writing.

Author Response

Dear Reviewer,

We have revised our paper according to the comments. Below please find the responses to the reviewers’ comments (in red) and details about the revisions made – changes in the paper are marked in blue in the revised version we submitted. Once again, thank you for your consideration.

Your sincerely,

Jurairat and Seng

Reviewer 2 Report

The contribution of the paper is very interesting from the point of view of various research cases, however it needs some development.

More adequate mathematical descriptions should be included in the paper. The paper is slightly chaotic - it makes impression that too much text and to colloquial expression are given. It needs proper mathematical apparatus (especially at the beginning of the paper), consistent analysis, result-effect sequence.

Are authors sure that density of nodes is given in correct form?

Algorithms should be referenced in the text of the manuscript before they are placed.

What type of probability distribution is used in RXW, etc.? It is important to mention it in the paper.

What are the assumptions for simulation model? Why there is no conceptual model mentioned? For this purpose Authors can follow the paper: https://doi.org/10.3390/e22040423

The construction of the paper is not classical. For example at the beginning there is lack of literature review and it is described at the section 5. Since the paper is connected to crowdfunding, it is worth mentioning similar methods well described in the paper: Czwajda L., Kosacka-Olejnik M., Kudelska I., Kostrzewski M., Sethanan K., Pitakaso R., 2019, Application of prediction markets phenomenon as decision support instrument in vehicle recycling sector, LogForum, Vol. 15, Issue 2, pp. 265-278. DOI: 10.17270/J.LOG.2019.329

Sensitivity analysis should be taken into consideration.

Conclusions seem like an abstract - it is suggested to present more general results here and moreover to mention potential future research.

Author Response

(The authors gave the same response as above.)

Round 2

Reviewer 2 Report

Uniform probability distribution is one of the most often used in this type of research. It does not mean that it is the best fitted. There is a study of such probability distribution usage in the paper: https://doi.org/10.3390/e22040423

It is suggested Authors to prepare similar discussion in their papers. It would make the paper more significant, especially that in the previous version of manuscript Authors completely omitted information about any probability distribution. And it is important to present it as precise and detailed as possible. This is the strong potential of this research - to analyse probability distributions used and the potential other one in future research. Obviously it is Authors decision, however it is strongly recommended to do so.

Similar methods to crowdsourcing were omitted.

Websites are given without accessed dates specifications.

Author Response

Dear Reviewer,

We have revised our paper according to the comments. Below please find the responses to the reviewers’ comments (in red) and details about the revisions made – main changes in the paper are marked in red in the 2nd revised version, we submitted and marked in blue in the 1st revised version. Once again, thank you for your consideration.

Your sincerely,

Jurairat and Seng
